# Effects of Charcoal Rot on Soybean Seed Composition in Soybean Genotypes That Differ in Charcoal Rot Resistance under Irrigated and Non-Irrigated Conditions

**DOI:** 10.3390/plants10091801

**Published:** 2021-08-29

**Authors:** Nacer Bellaloui, Alemu Mengistu, James R. Smith, Hamed K. Abbas, Cesare Accinelli, W. Thomas Shier

**Affiliations:** 1Crop Genetics Research Unit, USDA, Agricultural Research Service, 141 Experiment Station Road, Stoneville, MS 38776, USA; rusty.smith@usda.gov; 2Crop Genetics Research Unit, USDA, Agricultural Research Service, Jackson, TN 38301, USA; alemu.mengistu@usda.gov; 3Biological Control of Pests Research Unit, USDA, Agricultural Research Service, 59 Lee Road, Stoneville, MS 38776, USA; hamed.abbas@usda.gov; 4Department of Agricultural and Food Sciences, Alma Mater Studiorum, University of Bologna, Viale Fanin 44, 40127 Bologna, Italy; cesare.accinelli@unibo.it; 5Department of Medicinal Chemistry, College of Pharmacy, University of Minnesota, 308 Harvard Street, SE, Minneapolis, MN 55455, USA; shier001@umn.edu

**Keywords:** charcoal rot, soybean nutrition, soybean protein, soybean oil, soybean fatty oil, *Macrophomina phaseolina*

## Abstract

Charcoal rot is a major disease of soybean (*Glycine max*) caused by *Macrophomina phaseolina* and results in significant loss in yield and seed quality. The effects of charcoal rot on seed composition (seed protein, oil, and fatty acids), a component of seed quality, is not well understood. Therefore, the objective of this research was to investigate the impact of charcoal rot on seed protein, oil, and fatty acids in different soybean genotypes differing in their charcoal rot susceptibility under irrigated and non-irrigated conditions. Two field experiments were conducted in 2012 and 2013 in Jackson, TN, USA. Thirteen genotypes differing in charcoal rot resistance (moderately resistant and susceptible) were evaluated. Under non-irrigated conditions, moderately resistant genotypes showed either no change or increased protein and oleic acid but had lower linolenic acid. Under non-irrigated conditions, most of the susceptible genotypes showed lower protein and linolenic acid but higher oleic acid. Most of the moderately resistant genotypes had higher protein than susceptible genotypes under irrigated and non-irrigated conditions but lower oil than susceptible genotypes. The different responses among genotypes for protein, oil, oleic acid, and linolenic acid observed in each year may be due to both genotype tolerance to drought and environmental conditions, especially heat differences in each year (2012 was warmer than 2013). This research showed that the increases in protein and oleic acid and the decrease in linolenic acid may be a possible physiological mechanism underlying the plant’s responses to the charcoal rot infection. This research further helps scientists understand the impact of irrigated and non-irrigated conditions on seed nutrition changes, using resistant and susceptible genotypes.

## 1. Introduction

Soybean is a major crop in the world that is an important source of protein, oil, fatty acids, amino acids, sugars, minerals (seed composition) [1,2,3,4], and charcoal rot (*Macrophomina phaseolina*) causes significant yield loss to it [5]. Therefore, maintaining these seed nutrients or enhancing them is essential for human nutrition and livestock health. Although seed composition nutrients are genetically controlled, they are significantly affected by biotic (such as diseases) and abiotic (such as soil conditions, heat, and drought) stresses.

Charcoal rot is a major disease, causing soybean yield loss and reduced soybean seed quality [6], altering the level and profile of seed composition nutrients [7,8]. Charcoal rot is found throughout the north central and southern regions of the United States [9,10,11] as well as in tropical and subtropical regions of the world [9,11,12,13]. If the disease is severe, both yield and seed quality are reduced [13]. The disease infects at least 500 plant species, including crops such as corn, sorghum, cotton, tobacco, and soybean [12,14]. In spite of enormous efforts to identify charcoal rot resistance in soybean, no commercial soybean cultivars resistant to charcoal are available to growers. Others evaluated MG IV genotypes DT97-4290 [15,16], moderately resistant to *M. phaseolina,* and Egyptian, susceptible to *M. phaseolina* and MG III genotypes AG 3905, moderately resistant to *M. phaseolina*, and DK 3964, susceptible to *M. phaseolina*. They evaluated the disease at V5 (vegetative), R1 (beginning flowering), R3 (beginning pod set), R5 (beginning seed-fill), R6 (full seed-fill), and R7 (yellow pod color/physiological maturity) growth stages [17]. They found that the yield loss due to charcoal rot ranged from 6% to 33% under irrigated conditions and higher under non-irrigated conditions. They concluded that yield loss depended on disease severity, irrigation, and environmental conditions.

Information on the effects of charcoal rot on seed composition (protein, oil, and fatty acids) is limited, and what is available is still controversial [7,8,18]. For example, Bellaloui et al. [7] investigated the effects of charcoal rot infestation on seed composition under irrigated and non-irrigated conditions in DT97-4290, and Egyptian and Pharaoh (susceptible cultivars) [7,15]. They found that there were no significant differences in seed protein content in the moderately resistant germplasm line DT97-4290 under these conditions. However, protein content was significantly higher for the susceptible cultivars Egyptian and Pharaoh when irrigated and when non-infested relative to infested conditions. The seed content of oleic acid was significantly higher in susceptible cultivars under infested conditions than in seed of moderately resistant genotypes. The linolenic acid content was significantly lower in seeds of susceptible cultivars under infested conditions. They considered the decrease in protein and linolenic acid contents in Pharaoh under disease infestation to be part of the reduced seed quality associated with charcoal rot infection [19]. They concluded that charcoal rot can alter seed composition in susceptible cultivars, but its effects on moderately resistant will depend on the type of the seed composition nutrient. Since there was a lower protein content in Pharaoh under infested and non-irrigated conditions, the combination of infestation and non-irrigation may further lower the level of protein in seeds, consistent with the observation of others [20] who showed greater severity of charcoal rot infection under drought conditions. Furthermore, research on other diseases, such as the studies of [21] with *Sclerotinia sclerotiorum*, showed that increasing soybean infestation levels resulted in increased seed protein content in one cultivar, but not in four other cultivars studied. It was shown that soybean infected with bean pod mottle virus resulted in higher seed protein and lower oil levels [22]. The inverse relationship between protein and oil levels in the latter observation could be due to a genetically controlled relationship between nutrient components in seed [23,24,25]. Environmental interactions affecting genes and energy cost can significantly affect this inverse relationship between protein and oil syntheses [26] by unknown mechanisms.

Because changes and alterations in seed composition nutrients occur under biotic and abiotic stresses, and because the relationships between genotype and environment for seed composition are not well understood, the current research focuses on characterizing the effect of one of the major diseases (charcoal rot) and soil moisture (non-stressed and stressed) on seed protein, oil, and fatty acids. The current study used 13 genotypes, including breeding lines that are moderately resistant to charcoal rot and tolerant to drought [6], to examine the effects of charcoal rot on soybean seed protein, oil, and fatty acids under irrigated and non-irrigated conditions.

## 2. Results and Discussion

Thirteen genotypes differing in charcoal rot resistance (moderately resistant and susceptible) and drought tolerance were used (Table 1); extracted and modified from Mengitu et al. (2018 [6]. Colony forming units of *M. phaseolina* in the lower stem and root tissues at reproductive stage R7 were used to assess disease severity in irrigated and non-irrigated environments in 2012 and 2013 (Table 2); extracted and modified from Mengistu et al. (2018) [6]. Two irrigation treatments (irrigated and non-irrigated) were used. Irrigation water was controlled in each plot until the end of physiological maturity (R7 growth stage).

ANOVA showed that year and irrigation (Irr) had significant effects on protein, oil, and linoleic and linolenic acids (Table 3). Maturity (MG) significantly affected protein, oil, oleic, and linoleic and linolenic acids. Both drought tolerance and charcoal rot resistance (DT_R) significantly affected protein and oil. Genotype within MG by (DT_R) had significant effects on protein, oil, palmitic, and stearic, oleic, and linolenic acids. Genotype × year (MG × DT_R) also had significant effects on protein, oil, and linolenic acid. Since year was a factor that interacted with other factors for seed composition nutrients (Table 3), results are presented for each year.

In 2012, the moderately resistant and drought resistant genotypes showed either an increase or no change in protein (Table 4). The moderately resistant and drought resistant genotypes showed an increase in oleic acid and a decrease in linolenic acid under non-irrigation compared with under irrigation. A similar trend was shown in 2013 for protein, oleic, and linolenic acids in these genotypes, except that moderately resistant genotypes DS-880 and DT97-4290 generally exhibited no protein difference under irrigation and non-irrigation conditions. All moderately resistant genotypes showed higher seed oleic and lower linolenic under non-irrigation conditions. In 2013, and comparing genotypes under irrigation and non-irrigation conditions, the moderately resistant genotypes, including those that are drought resistant, showed either an increase or no change in protein (Table 5). Both moderately resistant genotypes DS-880 and DT97-4290 showed an increase in oleic acid and a decrease in linolenic acid under non-irrigation conditions. There was no clear trend for the remaining seed composition constituents. It is clear that seed protein, oil, and palmitic, stearic, oleic, linoleic, and linolenic acids differ in each genotype under irrigated vs. non-irrigated conditions. Comparing the level of charcoal rot infection between moderately resistance lines (MR) and susceptible controls (LS98-0358 and Pharaoh), MR showed resistance under irrigated and non-irrigated conditions in 2012, except for Dyna-Gro 36C44 (Table 2), above. In 2013, similar observations were made under irrigated conditions, except for R01-581F, and under non-irrigated conditions, except for Dyna-Gro 36C44 and R02-1325 (Table 2), above.

In 2012, significant negative correlations were observed between seed protein and stearic acid, palmitic and stearic acids, palmitic and oleic acids, and oleic and linoleic acids under irrigated and non-irrigation conditions (Table 6). A significant positive correlation was observed between seed palmitic and stearic acids under irrigated and non-irrigated conditions. Under irrigated conditions, there were positive correlations between seed palmitic and linolenic acids, between stearic and linolenic acids, and between linoleic and linolenic acids but a negative correlation between oleic acid and linolenic acid. Under non-irrigated conditions, there were negative correlations between seed oil and oleic acid and stearic and linolenic acid, but a positive correlation between stearic and linolenic acid. In 2013, significant negative correlations were observed between seed protein and oil, between stearic acid and protein, oleic and palmitic acids, palmitic and linoleic acids, oleic and linoleic acids, and oleic and linolenic acids (Table 7). Significant positive correlations between seed palmitic and stearic acids and palmitic and linolenic acids were observed. Positive correlations were observed between seed oil and stearic acid only under irrigated conditions.

However, significant positive correlation between oil and oleic acid was observed under non-irrigated conditions only. Negative correlations between seed oil and linoleic acid and oil and linolenic acid were observed under non-irrigated conditions only.

The non-change or increase of protein in moderately resistant genotypes, especially DS-880 and DT97-4290, and drought tolerant genotypes may indicate the ability of these genotypes to maintain the level of protein under drought conditions when charcoal rot infestation is high. Comparing moderately resistant (MR) with susceptible (S) genotypes, the average seed protein contents across all moderately resistant genotypes and across all susceptible genotypes were 41.6% (MR) and 40.34% (S) in 2012 and 41.6% (MR) and 39.8% (S) in 2013, supporting the observation that these genotypes had the ability to maintain higher protein content under irrigated and non-irrigated conditions. The increase of seed protein in MR genotypes across all genotypes in each year resulted in a decrease in oil in certain genotypes, supporting the inverse relationship between protein and oil [23,24,25]. The increase in seed oleic acid and decrease in linolenic acid in moderately resistant and drought resistant genotypes may be a possible mechanism for responding to drought and high heat stress. Previous research showed that charcoal rot infestation under irrigated and non-irrigated conditions resulted in no seed protein changes in DT97-4290 (moderately resistant to charcoal rot) compared with Egyptian and Pharaoh (susceptible to charcoal rot), whereas protein content was significantly lower under non-irrigated and infested conditions. They also found that seed oleic acid was significantly higher and linolenic acid was lower under infested and non-irrigated (drought) conditions. The reduction in seed quality reflected in seed protein and linolenic acid contents in the susceptible cultivar, Pharaoh, under infested and drought conditions could be due to a combination of the two stress forms [19]. It was reported that a combination of higher severity of charcoal rot and non-irrigated (drought) conditions may further lower the level of protein in seeds, supporting the observation of Kendig et al. [20]. Mengistu et al. [6] showed that disease severity, measured using average, median, and maximum CFU g^−1^ at R7, was greater in the non-irrigated environment compared to the irrigated environment in each year, indicating that charcoal rot infestation of soil and infection of plants occurred, and consequently, differences in seed composition constituents could be due to charcoal rot disease [7]. Others reported that increasing infection with *S. sclerotiorum* resulted in higher seed protein content in one cultivar and not in four other cultivars [21]. Ziems and colleagues [22] reported that soybean infected with bean pod mottle virus had higher seed protein and lower oil. Elevated seed protein and reduced oil in breeding lines and cultivars may be due to the gene-based inverse relationship between protein and oil [23,24,25], although this inverse relationship between protein and oil can be significantly affected by interactions between the environment and gene and energy cost [26].

Under non-irrigated conditions, most susceptible genotypes to charcoal rot in the present study showed lower seed protein and higher oleic acid compared with irrigated conditions, but lower linolenic acid. Conversely, moderately resistant genotypes showed either no difference in seed protein content or higher protein and oleic acid but lower linolenic acid under non-irrigated conditions, supporting previous research conducted by others [20,27]. The existence of both traits (resistance to charcoal rot and drought tolerance) in a genotype will provide the genotype the ability to maintain the level of seed protein under charcoal rot infestation and drought conditions compared to susceptible genotypes. The consistent inverse correlations between protein and oil, protein and stearic acid, oleic and linolenic acids, oleic and linolenic acids, oleic and palmitic acids, and the positive correlations between palmitic and stearic acids and palmitic and linolenic acids may be due to the role of desaturase enzymes in fatty acid biosynthesis in controlling downstream production of fatty acids from palmitic acid in the sequence palmitate → stearate → oleate → linoleate → linolenate. The desaturase enzyme activities were shown to be influenced by environmental factors (such as drought and heat) and biotic factors (such as diseases and genotypes). For example, it was shown that growing soybean at elevated temperatures resulted in a decrease in linoleic and linolenic acid concentrations in seed triacylglycerols and an increase in oleic acid [28,29,30]. The involvement of desaturase enzyme activity and their roles in environmental stress factors such as drought and high temperature and disease tolerance has also been reported in other studies [31,32,33,34]. It was reported that the relative quantities of saturated (palmitic and stearic) and unsaturated fatty acids (oleic, linoleic, and linolenic) are critical factors that influence the quality and commercial applications of plant oils (lower oleic acid and higher linolenic acid contribute to oil oxidation). It was concluded that understanding the mechanisms controlling fatty acid composition during seed development under varying environmental conditions is a critical step toward regulating processes involved in fatty acid production [35]. Different responses among genotypes for seed protein, oil, oleic acid, and linolenic acid in each year may be due to genotypic differences, maturity, and environmental conditions differences, especially heat and drought differences in each year, supporting previous research [7,26,36].

The present study is consistent with different accumulation of seed nutrients in genotypes in each year being due to temperature and drought. For example, the monthly average maximum air temperature in 2012 was 32, 34, and 33 °C for June, July, and August, respectively. In 2013, the monthly average maximum air temperature was about the same (31 °C) for all three months [6], indicating that 2012 was hotter than 2013. The frequency of precipitation from planting to harvest (May through October of each year) was 48 and 62 days in 2012 and 2013, respectively, with 23% more days of precipitation during the growing season in 2013, indicating that 2013 received more precipitation than 2012. The mean soil temperature over the season at the 5.1 cm depth for irrigated plots was 3 °C more in 2012 than in 2013. However, for the non-irrigated treatment, the mean soil temperature was on average 1 and 2 °C higher than for the irrigated treatment at the same depth in 2012 and 2013, respectively. The mean soil temperatures in non-irrigated plots at the 10 cm depth were 1 and 3 °C lower than in non-irrigated plots at the 5.1 cm depth in 2012 and 2013, respectively. Soil water potential (water deficit) was −100 kPa or greater in 2012 in July and August, but in 2013 it only occurred in September, and the highest soil water potential of −120 kPa was observed in July of 2012 under non-irrigated conditions, indicating that 2012 was hotter and drier than 2013, which was probably responsible for different accumulation of nutrients in seed in each year. Generally, and based on our experience in irrigation scheduling for irrigated plots, soil water potential is kept at about −15 to −20 kPa, which represents the field water holding capacity. Soil water potential was monitored by using soil water potential sensors and read by soil moisture meter (Watermark Company, Inc., Janesville, WI, USA). The negative impact of high heat and drought on nutrients uptake, translocation, and assimilation, and their effects on metabolism and assimilates were previously reported [37].

Positive and negative correlations between nutrients were previously reported and thought to be dependent on growth conditions, genotype, and nutrient supply [37,38]. The processes of nutrient uptake, translocation, assimilation, and accumulation in seed and their relations with genetics and environment are still not understood [2,3,39]. Positive or negative correlations have been observed in other studies, and they have been mainly attributed to gene × environment interactions [2,39,40,41,42]. Changes to the correlation between seed composition nutrients by effects of genotypes and environments is still not yet understood and further research is needed to understand the mechanisms controlling these changes. The large distribution of nutrients across genotypes and environmental effects, especially in protein, oil, oleic, and linoleic acids content, reflects the genetic differences in the efficiency of uptake, assimilation, and accumulation mechanisms for the various nutrients. Normal and bimodal distributions of nutrients reflect the complexity of these relationships, which result in some genotypes accumulating more nutrients than others (Figure 1, Figure 2, Figure 3 and Figure 4).

## 3. Materials and Methods

### 3.1. Growth Conditions

A field experiment was conducted in 2012 and 2013 at the West Tennessee Research and Education Center at Jackson, TN, USA (35.65 N latitude, 89.51 W longitude). The field has a known history of high charcoal rot disease pressure. Details about the experimental conditions were published elsewhere [6]. Briefly, the field had been under no-till management with continuous soybean cropping for over 10 years prior to the initiation of this experiment. A week before planting, the field was divided into six grids, and 10 soil cores were collected per grid in a zigzag pattern to determine the colony forming units (CFUs) per gram of soil (CFU g^−1^) [6]. The CFU g^−1^ of soil of *M. phaseolina* ranged from 300 to 1000, and the inoculum distribution of *M. phaseolina* remained within the same range throughout the field in each year. Colony forming units of *M. phaseolina* in the lower stem and root tissues at reproductive stage R7 were used to assess disease severity [6] in irrigated and non-irrigated environments in 2012 and 2013, as shown in Table 2 above; extracted and modified from Mengistu et al. (2018) [6]. The planting dates were 9 May in 2012 and 14 May in 2013. Seeding rate of approximately 400,000 seeds ha^−1^ was used with a four-row Almaco plot planter (99 M Avenue, Nevada, IA, USA) equipped with John Deere XP row units (Deere and Company, 1 John Deere Place, Moline, IL, USA) set on a 76.2 cm row spacing in four rows 5.38 m long. Millet grain completely colonized with microsclerotia of *M. phaseolina* was used to infest soil following a previously described protocol [11]. It was applied using a standard 31 cell cone type seeder (Almaco 99 M Avenue, Nevada, IA, USA) at planting at a rate of 1.0 g m^−1^ to minimize plot-to-plot variation in soil inoculum levels. Thirteen genotypes differing in charcoal rot resistance (moderately resistant and susceptible) and drought tolerance were used (Table 1 above [6]); extracted and modified from Mengistu et al. (2018) [6]. Two irrigation treatments (irrigated and non-irrigated) were used. Independent irrigation water for each plot was controlled until each genotype reached the end of physiological maturity (R7 growth stage). To avoid moisture diffusion from an irrigated plot to a non-irrigated plot, a buffer strip of four rows were planted between each irrigated and non-irrigated plot. Irrigation was provided with a drip irrigation system.

### 3.2. Seed Protein, Oil, and Fatty Acids

Protein, oil, and fatty acid contents in mature soybean seeds from all treatments and genotypes were analyzed with a Diode Array Feed Analyzer AD 7200 (Perten, Springfield, IL, USA). Seeds were ground by a Laboratory Mill 3600 (Perten, Springfield, IL, USA) and approximately 25 g of seed were analyzed for protein, oil, and fatty acid contents according to Bellaloui et al. [2,7]. Calibration equations were initially developed by the University of Minnesota and upgraded by the Perten company using Perten’s Thermo Galactic Grams PLS IQ software. The calibration equations were established according to AOAC methods [43,44]. Protein and oil were expressed on a dry-matter basis, and fatty acids (palmitic, stearic, oleic, linoleic, and linolenic) were expressed on a total-oil basis.

### 3.3. Experimental Design and Statistical Analysis

The details of the experimental design were provided by Mengistu et al. [6]. The design was a randomized complete block with a split-split plot with four replications. Maturity groups (MG) were main plots, genotype within MG were sub-plot, and irrigation (Irr) was sub-sub-plot. Statistical analyses to evaluate the effects of year, Irr, and genotype within MG, drought tolerance, and charcoal rot resistance (DT-R) and their interactions were conducted using PROC GLIMMIX (SAS, SAS Institute, 2002–2010) according to [6]. Replicate and its interactions with major factors (Year, Genotype, Irr, and MG) were considered as random effects. Year, MG, genotype within MG, Irr, and their interactions were considered as fixed effects. Mean comparisons were conducted by Fisher’s Protected LSD test and the level of significance of *p* ≤ 0.05 was used in SAS [45]. Correlations were estimated using PROC CORR in SAS.

## 4. Conclusions

The present study demonstrates that a combination of charcoal rot infestation in the soil, which results in the infection of soybean plants, and drought (no irrigation) can alter seed protein, oleic, linoleic, and linolenic acid contents. In moderately resistant genotypes, especially DS-880 and DT97-4290, and drought-tolerant genotypes, seed protein content either did not change or increased with no irrigation. However, seed oleic acid content increased, and linolenic acid decreased under the same conditions. Conversely, susceptible genotypes showed a decrease in seed protein and linolenic acid contents under non-irrigated conditions and lower oleic acid and higher linolenic acid under irrigation. It was concluded that moderately resistant and drought tolerant genotypes may have the ability to maintain the level of seed protein under non-irrigated conditions compared to susceptible genotypes. Seed oleic acid may be useful as a biochemical marker under drought and charcoal rot infection conditions. The positive and negative correlations between seed fatty acids (palmitic, stearic, oleic, linoleic, and linolenic) need further research because the controlling mechanisms regulating these relationships are complex and poorly understood. Specifically, the production of these fatty acids results from the action of desaturase enzymes, which are influenced by environmental factors, including temperature, drought, and diseases. Oleic acid is a mono-unsaturated fatty acid with one double bond. Apparently, under stress, plants are accumulating monounsaturated fatty acids versus polyunsaturated fatty acids (more than one double bond). Because the involvement of fatty acids is essential in the synthesis, stability, and integrity of plant cell membranes, and in seed oil quality, research is currently underway in the areas of physiology, genetics, and molecular biology to understand the mechanisms controlling seed oil production and quality.

## Figures and Tables

**Figure 1 plants-10-01801-f001:**
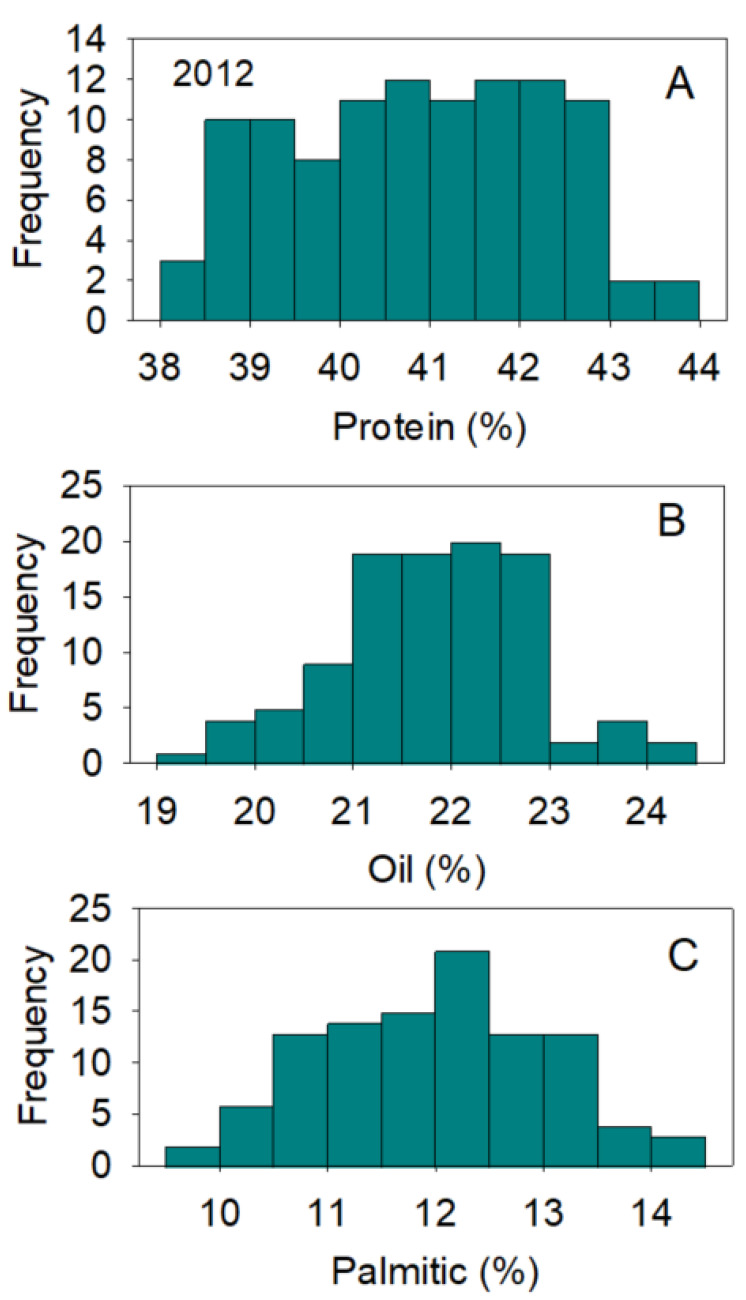
Distributions of soybean seed protein (**A**), oil (**B**), and palmitic acid (**C**) (%) across genotypes. The experiment was conducted in 2012 in Jackson, TN, USA. Gaps that exist in any distribution graph indicate there is zero line in that range. Frequency (Y-axis) refers to number of individual replicates of genotypes.

**Figure 2 plants-10-01801-f002:**
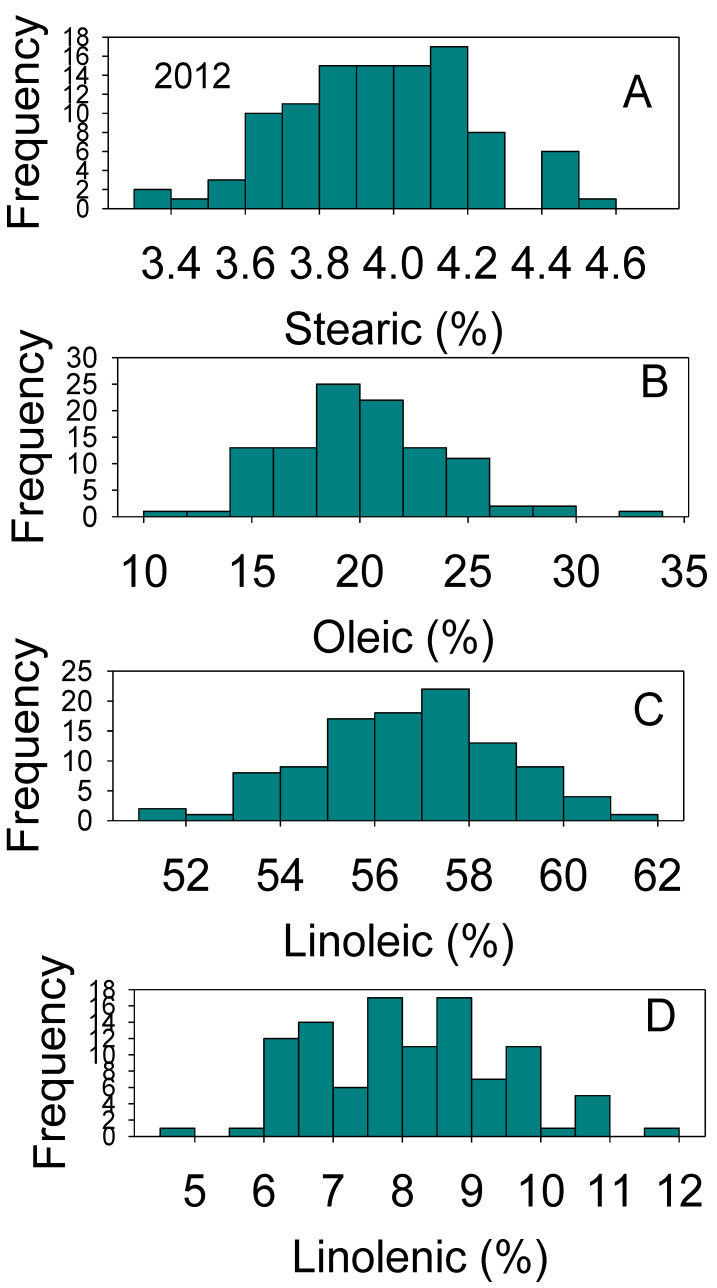
Distributions of soybean seed stearic (**A**), oleic (**B**), linolenic (**C**), and linolenic (**D**) acids content (%) across genotypes. The experiment was conducted in 2012 in Jackson, TN, USA. Gaps that exist in any distribution graph indicate there is zero line in that range. Frequency (Y-axis) refers to number of individual replicates of genotypes.

**Figure 3 plants-10-01801-f003:**
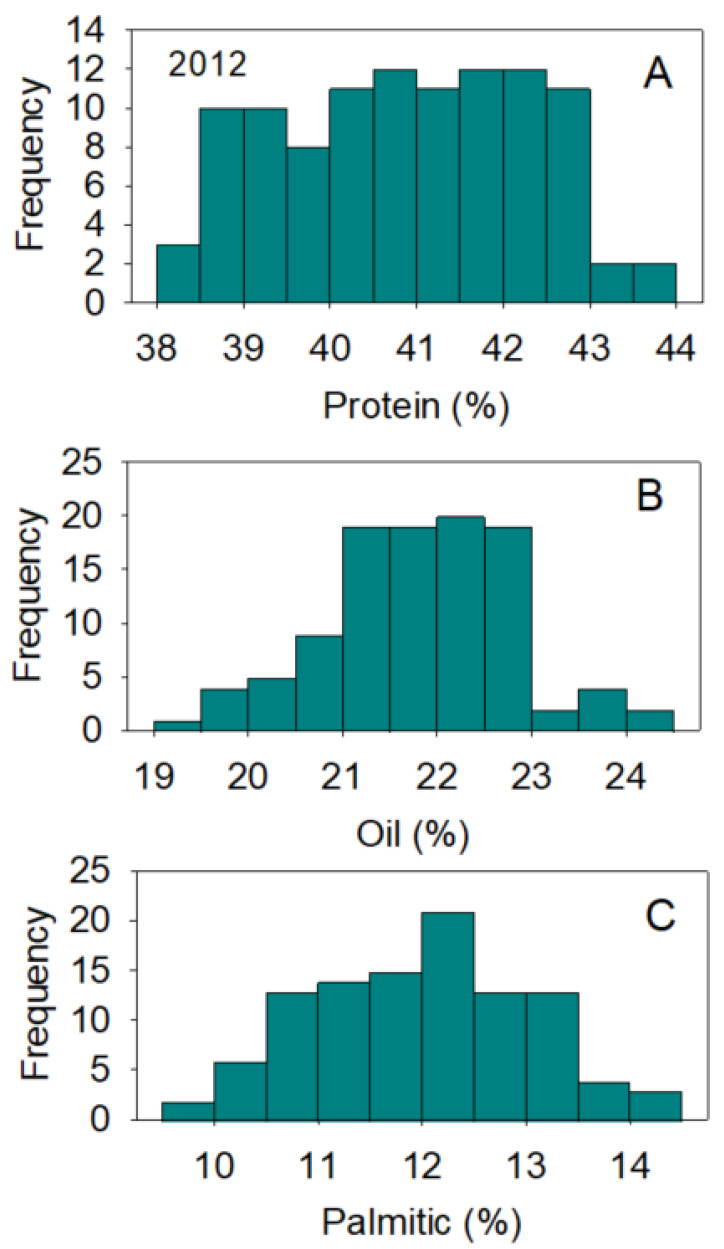
Distributions of soybean seed protein (**A**), oil (**B**), and palmitic acid (**C**) content (%) across genotypes. The experiment was conducted in 2013 in Jackson, TN, USA. Gaps that exist in any distribution graph indicate there is zero line in that range. Frequency (Y-axis) refers to number of individual replicates of genotypes.

**Figure 4 plants-10-01801-f004:**
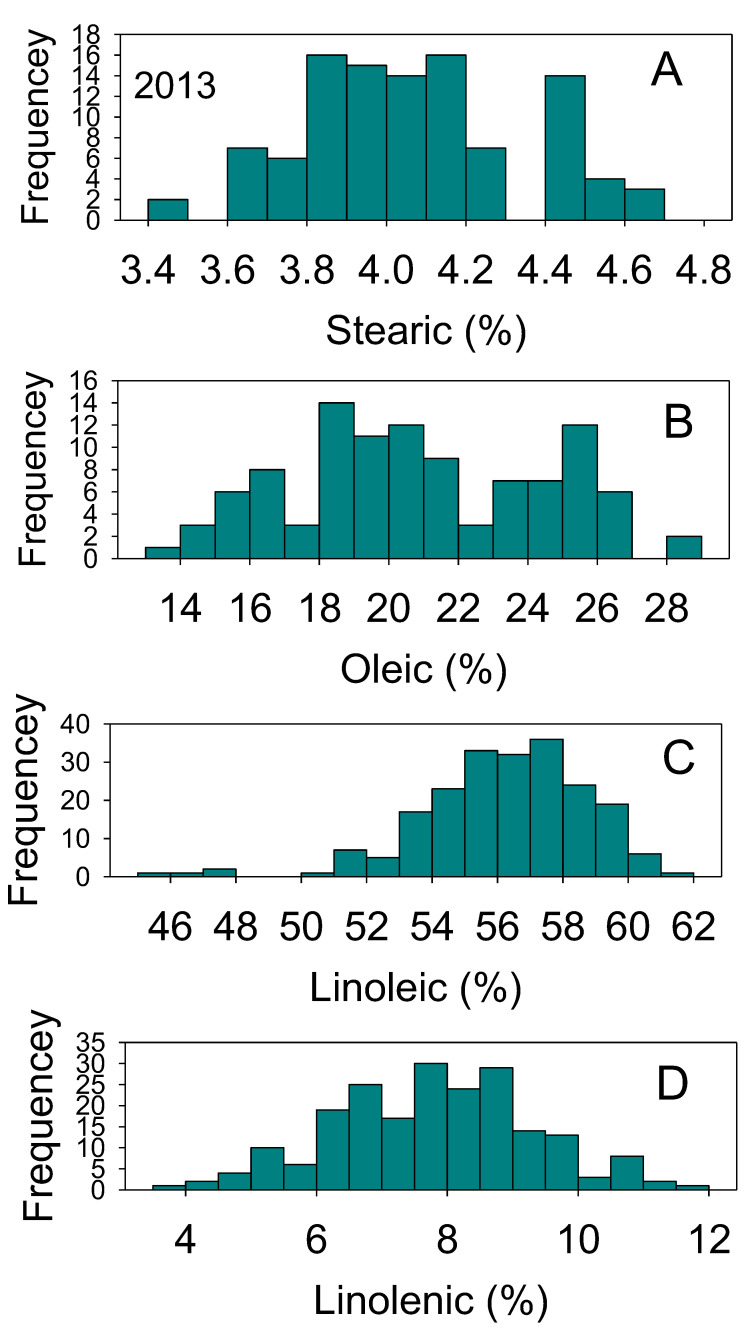
Distributions of soybean seed stearic (**A**), oleic (**B**), linolenic (**C**), and linolenic (**D**) acids content (%) across genotypes. The experiment was conducted in 2013 in Jackson, TN, USA. Gaps that exist in any distribution graph indicate there is zero line in that range. Frequency (Y-axis) refers to number of individual replicates of genotypes.

**Table 1 plants-10-01801-t001:** Drought tolerance and charcoal rot resistance properties of the thirteen soybean genotypes included in this study, of which six are maturity group (MG) IV and seven are MG V genotypes.

Genotype	Maturity Group	Drought Tolerant/Susceptible	Resistance/Susceptibility to Charcoal rot
DS-880	MG V	Unknown	Moderately resistant
DT97-4290	MG IV	Unknown	Moderately resistant
R07-7232	MG V	Tolerant	Moderately resistant
USG-75Z38	MG V	Tolerant	Moderately resistant
USG-Allen	MG V	Tolerant	Moderately resistant
Osage	MG V	Tolerant	Moderately resistant
Dyna-Gro36C44	MG IV	Tolerant	Susceptible
Progeny 4408	MG IV	Tolerant	Susceptible
R01-581F	MG V	Tolerant	Susceptible
R02-1325	MG V	Tolerant	Susceptible
Trisoy-4788	MG IV	Tolerant	Susceptible
LS98-0358	MG IV	Unknown	Susceptible
Pharaoh	MG IV	Unknown	Susceptible

Extracted and modified from Mengistu et al. (2018) [6].

**Table 2 plants-10-01801-t002:** *Macrophomina phaseolina* infection levels (colony forming units, CFU g^−1^) determined at growth stage R7 under irrigated and non-irrigated conditions for the thirteen soybean genotypes in 2012 and 2013.

	2012				2013			
Genotype	Irrigated		Non-Irrigated		Irrigated		Non-Irrigated	
DS-880	1572	DC	722	GF	1409	E	3175	DC
DT97-4290	212	E	590	G	1350	E	1278	D
R07-7232	1364	DC	1818	EGF	1976	E	5731	C
USG 75Z38	1158	EDC	2459	EDF	1643	E	3062	DC
USG Allen	567	ED	986	GF	2257	ED	3826	C
Osage	2249	BDC	4787	EDC	1953	E	3833	C
Dyna-Gro 36C44	11,427	BA	39,704	BA	7555	BC	43,326	A
Progeny 4408	1171	EDC	10,770	BC	6206	BCD	24,428	BA
R01-581F	5195	BAC	10,431	BC	25,879	A	30,004	BA
R02-1325	3634	BAC	12,694	BAC	3491	ECD	39,340	A
Trisoy 4788	3104	BDC	9438	DC	3730	ECD	15,142	B
LS98-0358	6635	BAC	37,097	BA	15,528	BA	42,328	A
Pharaoh	18,905	A	43,547	A	17,633	BA	32,183	BA

Letters that differ from each other in each column are significantly different at *p* ≤ 0.05. Extracted and modified from Mengistu et al. (2018) [6].

**Table 3 plants-10-01801-t003:** Analysis of variance (*p* level) for the effect of year, irrigation (Irr: irrigation and non-irrigation), maturity group (MG), drought and charcoal rot infestation (DT_S), genotype, and their interactions on soybean seed protein, oil, and fatty acids contents (%). The experiment was conducted in 2012 and 2013 in Jackson, TN, USA.

Effect	Protein	Oil	Palmitic	Stearic	Oleic	Linoleic	Linolenic
	*p* Level	*p* Level	(C16:0) *p* Level	(C18:0) *p* Level	C18:1 *p* Level	(C18:2) *p* Level	(C18:3) *p* Level
Year	*	***	**	**	ns	*	*
Irr	*	*	ns	***	***	**	***
Irr × Year	ns	***	ns	ns	ns	ns	ns
MG	***	***	ns	*	***	***	***
MG × Year	*	*	ns	ns	ns	ns	ns
MG × Irr	ns	ns	ns	ns	ns	*	ns
MG × Irr × Year	ns	ns	ns	ns	ns	ns	ns
DT_S	***	*	ns	***	*	***	ns
DT-S × Year	**	*	***	***	ns	ns	ns
DT-S × Irr	***	*	ns	ns	ns	ns	ns
DT-S × Irr × Year	ns	*	ns	*	ns	ns	ns
MG × DT-S	ns	***	ns	ns	ns	ns	***
MG × DT-S × Year	ns	ns	ns	ns	ns	ns	**
MG × DT-S × Irr	ns	ns	ns	ns	*	ns	ns
MG * DT-S * Irr * Year	ns	ns	ns	ns	ns	ns	ns
Genotype (MG × DT-S)	***	***	***	***	***	***	***
Genotype × Year (MG × DT-S)	*	**	ns	ns	***	ns	**
Genotype × Irr (MG × DT-S)	***	ns	ns	ns	ns	ns	ns
Genotype × Irr × Year (MG × DT-S)	ns	ns	ns	ns	ns	ns	ns
Residuals	0.75	0.66	1.19	0.03	6.55	4.54	1.04

* Significance at *p* ≤ 0.05; ** significance at *p* ≤ 0.01; *** significance at *p* ≤ 0.001; ns, not significant.

**Table 4 plants-10-01801-t004:** Effects of irrigation (irrigated and non-irrigated) on soybean seed protein, oil, and fatty acids (palmitic, stearic, oleic, linoleic, and linolenic) contents (%) in susceptible (S) and moderately resistant (MR) genotypes. The experiment was conducted in 2012 in Jackson, TN, USA.

				**Irrigated**	**2012**			
**Genotype**	**Resistance**	**Protein**	**Oil**	**Palmitic**	**Stearic**	**Oleic**	**Linoleic**	**Linolenic**
DS-880	MR	42.15	22.20	12.50	4.08	17.35	56.58	9.40
DT97-4290	MR	42.43	21.95	11.90	4.00	21.98	54.95	8.03
R07-7232	MR	40.55	21.00	12.88	3.90	15.48	58.30	9.43
USG-75Z38	MR	40.93	21.78	11.63	3.90	20.50	56.48	8.35
USG-Allen	MR	40.68	22.35	11.48	3.70	15.08	60.43	9.68
Osage	MR	42.68	20.58	10.88	3.73	20.88	57.20	7.10
Dyna-Gro36C44	S	40.60	22.18	12.18	4.05	21.90	54.53	8.33
Progeny 4408	S	39.60	22.28	11.78	4.13	19.93	56.25	7.38
R01-581F	S	39.40	21.73	11.90	4.18	16.93	58.53	8.25
R02-1325	S	38.85	21.85	12.43	4.10	16.65	57.18	9.45
Trisoy 4788	S	40.03	22.90	12.58	4.23	17.50	57.28	9.13
LS98-0358	S	41.83	22.65	10.93	3.78	19.73	56.53	9.13
Pharaoh	S	42.05	21.88	10.98	3.78	19.65	58.43	9.05
LSD		0.43	0.34	0.46	0.11	1.29	0.77	0.57
				**Non-Irrigated**	**2012**			
**Genotype**	**Resistance**	**Protein**	**Oil**	**Palmitic**	**Stearic**	**Oleic**	**Linoleic**	**Linolenic**
DS-880	MR	42.23	22.08	12.83	4.15	21.43	56.28	8.13
DT97-4290	MR	42.00	21.28	12.45	4.25	23.50	54.60	6.73
R07-7232	MR	40.78	20.80	13.00	3.85	18.35	57.28	8.18
USG-75Z38	MR	41.05	21.15	11.50	3.90	24.23	57.38	6.63
USG-Allen	MR	42.25	21.45	12.20	3.80	18.68	57.65	8.25
Osage	MR	41.45	20.18	11.58	3.95	21.43	57.08	6.93
Dyna-Gro36C44	S	39.03	21.63	10.90	4.13	24.98	55.18	7.60
Progeny 4408	S	40.55	22.43	11.08	4.08	26.18	53.20	6.30
R01-581F	S	39.23	22.08	12.10	4.13	19.35	57.30	7.08
R02-1325	S	39.55	21.10	12.75	4.15	18.13	57.15	7.80
Trisoy 4788	S	39.08	23.53	12.68	4.20	21.13	54.73	7.88
LS98-0358	S	40.18	22.28	11.93	4.08	22.00	56.10	7.38
Pharaoh	S	40.98	21.43	11.70	4.03	20.00	57.03	7.60
LSD		0.40	0.39	0.46	0.09	1.37	0.78	0.47

LSD, least significant difference test; significant at *p* ≤ 0.05. Within each column, the difference between two values is statistically significant if it equals or exceeds the corresponding LSD.

**Table 5 plants-10-01801-t005:** Effects of irrigation (irrigation and non-irrigation) on soybean seed protein, oil, and fatty acids (palmitic, stearic, oleic, linoleic, and linolenic) contents (%) in susceptible (S) and moderately resistant (MR) genotypes. The experiment was conducted in 2013 in Jackson, TN, USA.

				**Irrigated**	**2013**			
**Genotype**	**Resistance**	**Protein**	**Oil**	**Palmitic**	**Stearic**	**Oleic**	**Linoleic**	**Linolenic**
DS-880	MR	40.98	20.08	11.63	4.00	20.68	55.13	9.50
DT97-4290	MR	40.90	20.20	11.83	4.00	20.88	55.53	6.65
R07-7232	MR	40.78	21.33	12.78	3.95	21.30	53.43	9.50
USG-75Z38	MR	40.78	20.58	10.48	3.85	22.38	56.48	6.53
USG-Allen	MR	39.90	19.43	11.65	3.83	17.45	59.95	8.83
Osage	MR	43.68	18.78	11.83	3.75	19.00	57.10	7.88
Dyna-Gro36C44	S	41.18	21.95	12.25	4.13	20.73	54.03	8.20
Progeny 4408	S	38.45	21.65	13.00	4.38	18.18	55.68	7.80
R01-581F	S	39.35	21.20	13.13	4.13	18.98	55.85	7.20
R02-1325	S	38.78	21.18	11.83	3.95	22.68	53.93	8.25
Trisoy 4788	S	39.90	22.93	12.05	4.05	22.75	55.65	6.15
LS98-0358	S	40.75	22.68	13.50	4.30	17.65	57.08	8.45
Pharaoh	S	41.28	19.53	12.33	4.23	18.33	57.18	7.23
LSD		0.56	0.38	0.59	0.08	1.63	1.42	0.65
				**Non-Irrigated**	**2013**			
**Genotypes**	**Resistance**	**Protein**	**Oil**	**Palmitic**	**Stearic**	**Oleic**	**Linoleic**	**Linolenic**
DS-880	MR	42.10	20.55	13.00	4.10	23.75	54.35	8.40
DT97-4290	MR	40.83	20.65	12.98	4.23	24.38	53.33	5.70
R07-7232	MR	42.25	20.75	12.55	3.95	27.85	58.00	8.78
USG-75Z38	MR	42.25	20.95	11.50	3.93	26.63	54.73	5.80
USG-Allen	MR	41.68	20.28	11.80	3.88	20.80	58.90	6.90
Osage	MR	41.08	20.35	12.28	3.98	19.18	57.10	7.03
Dyna-Gro36C44	S	39.83	21.85	12.10	4.30	23.70	53.63	7.85
Progeny 4408	S	39.55	21.80	12.95	4.43	23.63	54.13	6.05
R01-581F	S	39.78	20.83	12.78	4.30	18.70	57.78	7.10
R02-1325	S	40.13	20.50	12.93	4.25	17.98	55.43	8.18
Trisoy 4788	S	39.58	22.98	13.33	4.43	25.18	51.43	5.45
LS98-0358	S	39.58	22.63	13.38	4.20	19.90	54.68	7.83
Pharaoh	S	39.40	21.78	12.88	4.28	21.15	55.78	5.88
LSD		0.48	0.53	0.58	0.12	1.21	1.10	0.61

LSD, least significant difference test; significant at *p* ≤ 0.05. Within each column, the difference between two values is statistically significant if it equals or exceeds the corresponding LSD.

**Table 6 plants-10-01801-t006:** Pearson correlation coefficients (r and *p* values) between seed protein, oil, and fatty acids (palmitic, stearic, oleic, linoleic, and linolenic) contents (%). The experiment was conducted in 2012 in Jackson, TN, USA.

			**IRR Year 2012**				
**Nutrient**	**Protein**	**Oil**	**Palmitic**	**Stearic**	**Oleic**	**Linoleic**	
Oil	ns						

Palmitic	*r* = −0.29	ns					
	*p* = *						
Stearic	−0.47	ns	0.59	ns			
	***		***				
Oleic	0.36	ns	−0.44	ns			
	**		***				
Linoleic	ns	ns	ns	ns	−0.64		
					***		
Linolenic	ns	ns	0.40	0.32	−0.60	0.28	
			**	*	***	*	
			**NIRR Year 2012**				
**Nutrient**	**Protein**	**Oil**	**Palmitic**	**Stearic**	**Oleic**	**Linoleic**	**Linolenic**
Oil	*r* = −0.48						
	*p* = ***						
Palmitic	ns	ns					

Stearic	−0.29	ns	0.34				
	*		**				
Oleic	ns	0.26	−0.6384				
		*	***				
Linoleic	ns	−0.44			−0.52		
		**			***		
Linolenic	ns	ns	ns	−0.32	ns	ns	
				*			

* Significance at *p* ≤ 0.05; ** significance at *p* ≤ 0.01; *** significance at *p* ≤ 0.001. ns, not significant; IRR, irrigated; NIRR, non-irrigated. In each column and in each cell, the top value is r and the bottom value is *p*.

**Table 7 plants-10-01801-t007:** Pearson correlation coefficients (*r* and *p* values) between seed protein, oil, and fatty acids (palmitic, stearic, oleic, linoleic, and linolenic)contents (%). The experiment was conducted in 2013 in Jackson, TN, USA.

			**IRR 2013**			
**Nutrient**	**Protein**	**Oil**	**Palmitic**	**Stearic**	**Oleic**	**Linoleic**
Oil	*r* = −0.3757					
	*p* = ***					
Palmitic	ns	ns				

Stearic	−0.29	0.38	0.53			
	*	***	***			
Oleic	ns	ns	−0.40	ns		
			***			
Linoleic	ns	ns	ns	ns	−0.56	
					***	
Linolenic	ns	ns	0.49	ns	−0.28	ns
			***		*	
			**NIRR Year 2013**			
**Nutrient**	**Protein**	**Oil**	**Palmitic**	**Stearic**	**Oleic**	**Linoleic**
Oil	*r* = −0.38					
	*p* = ***					
Palmitic	ns	ns				

Stearic	−0.39	ns	0.55			
	**		***			
Oleic	ns	0.34	−0.29	ns		
		**	*			
Linoleic	ns	−0.28	−0.32	−0.29	−0.59	
		*	*	*	***	
Linolenic	ns	−0.27	0.36	ns	−0.50	ns
		*	**		***	

* Significance at *p* ≤ 0.05; ** significance at *p* ≤ 0.01; *** significance at *p* ≤ 0.001. ns, not significant; IRR, irrigated; NIRR, non-irrigated. In each column and in each cell, the top value is r and the bottom value is *p*.

## Data Availability

Not applicable.

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
