# Peer review of "Effects of Charcoal Rot on Soybean Seed Composition in Soybean Genotypes That Differ in Charcoal Rot Resistance under Irrigated and Non-Irrigated Conditions"

_plants, 2021, doi:10.3390/plants10091801_

Round 1
Reviewer 1 Report
The authors reported the effects of charcoal rot on soybean seed protein, oil, and fatty acids composition in different genotypes and environmental conditions.
The manuscript is written in a very particular way, in the negative sense of the statement, because it is completely incomprehensible to a reader reading it for the first time. The only way to understand its meaning is to read the authors' previous article which explains the experimental plan, the levels of infection and the water status of the environments. I don't think it is the correct way to present the results even if they are strictly linked to a previous work. In my opinion the work must be completely rewritten.
Introduction:
Lines 46-49: this is the aim of the work, move to the end of the introduction.
Lines 58-71: this detailed explanation of a previous work is suitable for a review article, not for the introduction of an original article. Please reduce and report only essential results.
Line 94 and along the manuscript: avoid starting a sentence with: “[22] showed that soybean…”, use more elegant sentence such as: “Ziems and colleagues [22] showed that soybean…”.
Results:
Tables 1 and 2 are fundamental for understanding the experimental plan. These tables, on the other hand, were inserted at the beginning of the results without ever being cited in the text. Authors must begin the results with some basic information and clearly state the experimental plan, genotypes and environment.
In the experimental plan were compared drought resistance/tolerance with resistance/susceptibility to charcoal rot, but the distinction in the text is unclear. Authors must find a way to clearly express what they are talking about. For example, instead of “moderately resistant and drought resistant genotypes” at line 121, “moderately resistant to charcoal rot and drought resistant genotypes”. There are no drought sensitive genotypes in Table 1.
I suggest better explaining the tables and figures in the results. In addition to specifying what is significant or not, the authors should further analyze the links among soybean seed protein, oil, and fatty acids with genotype, environment, disease resistance and drought tolerance.
Reviewer 2 Report
Comments:
The paper “Effects of charcoal rot on soybean seed composition in soybean genotypes that differ in charcoal rot resistance under irrigated and non-irrigated conditions” by Bellaloui and colleagues is interesting and this is an important research to better understand the impact of charcoal rot on seed protein, oil, and fatty acids in different soybean genotypes differing in their charcoal rot susceptibility under irrigated and non-irrigated conditions.
The paper is clear, well written and well designed and organised.
It may merit publication in my opinion, although the data collected refer to field experiments that were implemented some years ago (in 2012 and 2013).
However, my major concern is this: there is a similar paper by Bellaloui, Mengistu and Paris (2008) where main results indicate that charcoal rot infection may alter seed composition and nitrogen fixation in soybean. Moreover, the authors of that study conclude that the alteration in seed composition in terms of proteins, oil and fatty acids in seeds and that those alterations depended on cultivar susceptibility to charcoal rot and irrigation management. See reference [7]. Please, prove me that this paper now you want to publish is innovative and deserves publication after 8-9 years of the collection of the data.
The abstract is clear, pointing out the main results.
The Introduction is well written, and the objectives are clearly stated.
Methods are appropriate and are well explained. Statistics methods are suitable. The images/figures and tables are appropriate and elucidative. However, the graphs should be improved in terms of quality, size of the scales and in terms of legends.
Also, I do not agree with the position in the manuscript of tables 1 and 2. They are referred in Methods section and they appear immediately at the end of the Introduction. Please try to correct this.
Is there a way to know how much water had been added to soil in the irrigation treatment?
Was the field capacity and the maximum water holding capacity of the soil in the two situations determined? I realize that you determined the Soil water potential (water deficit) in 2012 in July and August, but in 2013 it only occurred in September, and the highest soil water potential of -120 kPa was observed in July of 2012 under non-irrigated conditions, indicating that 2012 was hotter and drier than 2013.
Were some properties and parameters of the irrigation water determined? For example, conductivity, salt concentration, …
Results and Discussion:
Results are well described, with the appropriate number of tables and figures, well organised and are important to the state of the art of this field of responses of plants to environmental stresses.
I believe there is some text missing before line 186…
Discussion is very complete and well supported in the cited literature.
Conclusions are well supported by the results.
The list of References is very extensive and complete, but I am afraid that there are excessive self-citations (22% of the citations are from the same authors and co-authors)…
Some other Minor things:
A minor comment: if you refer L. as the author of the species Glycine max, please refer also the authors of the fungal species when these are cited the first time in the text (examples Macrophomina phaseolina.and Sclerotinia sclerotiorum).
Line 74: do not begin the sentence by “[7] investigated”. Replace by “Bellaloui and colleagues investigated…. [7, 15]”. Please proceed accordingly along the text: lines 92 and 94, and for example lines 211 and 217…
Round 2
Reviewer 1 Report
The authors improved the manuscript as required.
Reviewer 2 Report
Dear Authors,
Many thanks for your work in order to improve the manuscript. I believe now that the paper can be published, after all the improvements that you carried out. All my major doubts and concerns were properly adressed and answered. Thanks for this!
Best wishes!